Review

Subject Area:
cellular biology/biochemistry/molecular biology

Keywords:
p62, riboregulation, vault RNA 1-1

Author for correspondence:
Matthias W. Hentze
e-mail: hentze@embl.org

# 'High vault-age': non-coding RNA control of autophagy

Magdalena Büscher[1,2], Rastislav Horos[1] and Matthias W. Hentze[1]

[1]European Molecular Biology Laboratory, Meyerhofstrasse 1, 69117 Heidelberg, Germany
[2]Collaboration for joint PhD degree between EMBL and Heidelberg University, Faculty of Biosciences, Heidelberg, Germany

MB, 0000-0002-2215-3329; RH, 0000-0001-7730-9557; MWH, 0000-0002-4023-7876

RNA-binding proteins typically change the fate of RNA, such as stability, translation or processing. Conversely, we recently uncovered that the small non-coding vault RNA 1-1 (vtRNA1-1) directly binds to the autophagic receptor p62/SQSTM1 and changes the protein's function. We refer to this process as 'riboregulation'. Here, we discuss this newly uncovered vault RNA function against the background of three decades of vault RNA research. We highlight the vtRNA1-1-p62 interaction as an example of riboregulation of a key cellular process.

## 1. Vault RNAs—small, non-coding and mysterious

Even though vault RNAs have been discovered more than thirty years ago, the molecular function of these abundant, small non-coding RNA polymerase III (Pol III) transcripts has remained unclear [1,2]. Vault RNAs were initially described by Kedersha and Rome as components of 13 MDa ribonucleoprotein assemblies that were identified serendipitously while isolating coated vesicles from rat liver. The ovoid morphology and arch-like structure of these complexes reminiscent of gothic cathedral ceilings prompted their naming as 'vault particle' or 'vaults' [1].

Vaults are the largest ribonucleoprotein complexes known to date. They measure $400 \times 400 \times 700$ Å and hence could enclose cellular structures bigger than ribosomes [3,4]. Vaults can reach high copy numbers (10 000–100 000 per cell) in organisms ranging from protists to humans [5–10]. Their structure as well as protein composition are highly conserved, suggesting a fundamental function in eukaryotic cells [11]. The main constituent of the particle is the major vault protein (MVP, 99 kDa), which accounts for more than 70% of the particle mass [12] (reviewed in [13]). Structural studies revealed that the expression of MVP alone suffices for the assembly of vault-like nanoparticles [14,15]. However, full integrity and a morphology indistinguishable from tissue-derived vaults requires co-expression of the two minor vault proteins, vault poly-(adenosine diphosphate-ribose) polymerase (VPARP, 193 kDa) and telomerase-associated protein 1 (TEP1, 290 kDa) [16]. By contrast, the association of vault RNA at the caps of the vault particle does not alter the particle's general morphology [7,12,17,18]. Importantly, most of the vault RNA is not associated with the particle [17], implying that it could well be involved in additional cellular interactions. Like the vault particle, the detailed function and mechanism of action of vault RNAs has remained mysterious over decades.

## 2. The genomic organization of vault RNA genes

Two vault RNA loci are syntenically conserved across most mammals [11]. In humans, the VTRNA1 locus is situated between the ZMAT2 (zinc finger matrin-type 2) gene and the PCHD (protocadherin) cluster, while the VTRNA2 locus is found in close proximity, between TGFB1 (transforming growth factor beta 1) and SMAD5 (SMAD family member 5) on the same chromosome [11,19–21]. The vault RNA promoters exhibit considerable

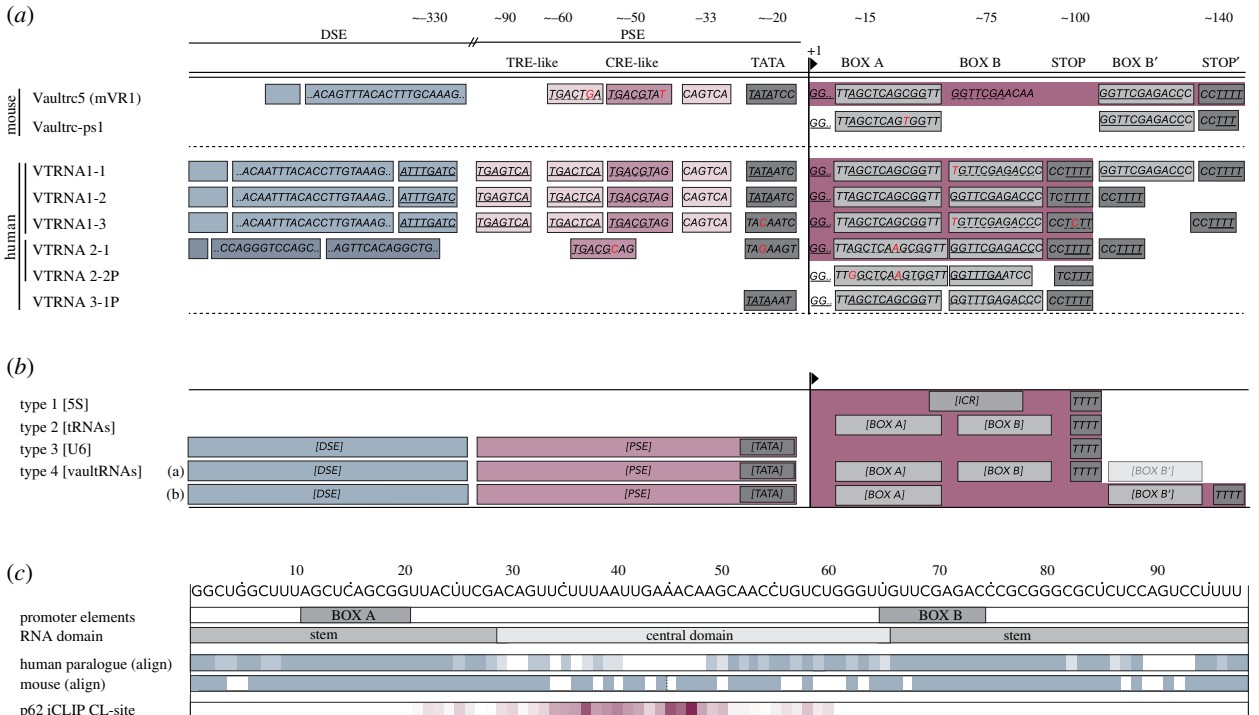

**Figure 1.** Vault RNAs are expressed from unusual RNA polymerase III (Pol III) promoters. (*a*) Overview of vault RNA loci. Transcriptional elements of the human and mouse vault RNA gene family (largely based on [11]). Depicted are key sequence elements of the vault RNA gene promoter and regulatory elements. The transcribed gene body is indicated by dark red background shading. Specific differences between sequence elements of the vault RNAs are highlighted in red. The name and location of sequence elements relative to the transcription start sites are indicated above. Underlined regions indicate canonical transcription factor-binding or termination motifs. (*b*) Different Pol III promoter types and their features. Polymerase III type 1, 2 and 3 promoters have been previously described [23]. The composite nature of vault Pol III type 4 promoters was initially proposed by [24]. (*c*) Vault RNA transcript features. Alignment of vault RNA features with its sequence. Top row, numbering from the transcription start site and sequence. Below, location of internal promoter elements within the transcript and structure predictions according to thermodynamic models. Middle, sequence alignment of human vtRNA1-1 to the human vault RNA paralogues or mouse mVR1 according to LocARNA (http://rna.informatik.uni-freiburg.de, v. (4.5.8); [25–27]. Darker shading represents increased conservation. Below, mean cross-link site values in p62 IPs according to individual nucleotide cross-link and immunoprecipitation (iCLIP). Darker shading represents increased cross-linking of p62 to vtRNA1-1 [28]. vtRNA, vault RNA; DSE, distal sequence element; PSE, proximal sequence element; CRE, cAMP responsive element; TRE, tetradecanoylphorbol acetate response element; TATA, TATA box element.

differences between the two syntenically conserved loci possibly resulting in differential expression patterns of the encoded vault RNAs [11]. So far, no functional relationship between vault RNAs and the syntenically conserved genes has been uncovered. However, with the newly described link between vault RNAs and autophagy (see below), it is noteworthy that members of the protocadherin family have also been described to associate with autophagy-related proteins and to influence lysosome targeting [22]. Additional clues that could explain the syntenic conservation might be uncovered in the future.

The substantial variation in length and sequence of the vault RNA central domain (figure 1) has hampered homology-based searches for vault RNAs in other species [11]. Experimental validation has so far been obtained for a single vault RNA in *M. musculus* and *R. norvegicus* [2,24,29], two in *R. catesbeiana* [2], four paralogues in *H. sapiens* [7,30] and one each in *S. purpuratus* [31], *T. brucei* [32], *D. rerio* and *O. latipes* [11]. Interestingly, in *S. purpuratus* the vault-associated RNA seems to contribute to vault particle integrity [31]. Furthermore, the development of iterative algorithms led to the identification of more than 100 potential vault RNA genes in deuterostome genomes [11]. Strikingly, the vault RNA 5′ and 3′ regions are predicted to form double-stranded structures in all species [2,11]. It has further been speculated that the function of the relatively long rodent vault RNA could encompass the function of several smaller RNA in other species [33].

The human genome encodes three vault RNA paralogues at the *VTRNA1* locus (*VTRNA1-1*, *VTRNA1-2* and *VTRNA1-3* (formerly *HVG-1/2/3* [7]) and one at the *VTRNA2* locus (*VTRNA2-1* (also referred to as pre-*miR-886* or *CBL3*) [19,20]), both located on chromosome 5q31. In addition, two vault RNA pseudogenes—*VTRNA2-2P* and *VTRNA3-1P* (formerly *HVG-4*)—are annotated in the human genome assembly hg38 on chromosomes 2 and X, respectively (figure 1*a*). The existence of more vault RNA pseudogenes without syntenic conservation has been proposed [11]. For the purpose of simplicity, we refer to the transcripts deriving from the two vault RNA loci as 'vtRNA1-1', 'vtRNA1-2', 'vtRNA1-3' and 'vtRNA2-1' in this review.

When comparing different species, the vault RNA genes show only limited sequence conservation beyond their internal Pol III type II promoter elements [11]. These comprise the box A and box B motifs, which are typically found in tRNA genes (figure 1) [2,11]. The two internal promoter elements serve as binding sites for the transcription factor TFIIIC, which in turn positions TFIIIB immediately upstream and thereby facilitates Pol III binding to the transcription start site (reviewed in [23]). Generic Pol III type II genes do not include additional upstream promoter sequences. However, mutational analysis of the rat vault RNA gene uncovered additional external promoter elements that contribute to transcription efficiency and regulation [24]. These elements include CRE-like (cAMP response element) and TRE-like (tetradecanoylphorbol acetate

response element) motifs as part of a proximal sequence element and further distal sequence elements (figure 1) [2,11]. The proximal CRE-like element is highly conserved between different species [11] and present upstream of all transcribed human paralogues, while it is missing for the vault RNA pseudogenes (figure 1), implying that it could be a determinant of vault RNA expression. CRE- and TRE-like elements are known to bind CREB and AP-1 transcription factor complexes, respectively, which integrate growth factor, nutrient and stress signalling—including bacterial and viral infections—to control key cellular processes such as proliferation, survival and differentiation [34–38]. In fact, the induction of vault RNA transcription upon viral infection [19,20,39,40] and the responsiveness of intracellular vtRNA1-1 levels to starvation [28] indicate that these transcription factor complexes could play a role in regulating vault RNA expression levels. In addition, NF-κB signalling and p65/RELA binding to the distal promoter region of *VTRNA1-1* were shown to promote its transcription upon viral infection [39]. Further experiments may determine synergies between, and determinants of, transcription factor binding at the vault RNA loci.

Human *VTRNA1-1* as well as other primate vault RNAs harbour a second copy of the box B motif and termination sequence downstream of the transcribed gene body that has been reported to negatively influence transcription depending on the upstream promoter sequences [11,24]. It is a feature that distinguishes the human *VTRNA1-1* gene from the other human vault RNA paralogues (figure 1).

Overall, the unique vault RNA Pol III promoter composition including type-2 internal sequences as well as type-3 upstream elements that act synergistically has been suggested to constitute a separate class of Pol III promoters (figure 1*b*; [24]). Since similar composite promoter arrangements are found in viral RNAs (e.g. EBER in EBV) and the protein MVP is able to self-assemble, the vault complex had been discussed to be an evolutionary relict of an early viral symbiont [20].

## 3. Expression of vault RNAs

The rat vault RNA shows uneven expression levels in different tissues, with particularly high abundance in the spleen, intestine and heart, and low levels in the brain, liver and kidney [2]. Similarly, the relative expression levels of the vault RNA paralogues vary in different human cell lines. However, vtRNA1-1 represents the predominant vault RNA species deriving from the *VTRNA1* locus in most cell lines examined [21]. A higher association of vtRNA1-3 with the vault particle has been reported in multi-drug-resistant cell lines independent of total vtRNA1-1 levels [21], suggesting that vtRNA1-3 is the prime RNA interacting with the vault particle in this context. However, the functional relevance and molecular details of this observation remain to be elucidated.

Besides sequence differences in the promoter region (figure 1) [11,21] and variation in the spacing of internal box A and box B elements [21], epigenetic modifications could contribute to the differential expression of the vault RNA paralogues. Promoter methylation was shown to inversely correlate with the expression levels of vtRNA1-2, vtRNA1-3 and vtRNA2-1 [41,42]. Interestingly, DNA hypermethylation, especially of the *VTRNA2-1* gene, further correlated with poor prognosis of some cancers, suggesting a potential role of this non-coding RNA as a tumour suppressor [41–46]. By contrast, the *VTRNA1-1* locus does not seem to be subject to methylation in a similar fashion [41]. The proximal promoter regions of all four expressed vault RNA paralogues are nucleosome depleted, facilitating active transcription initiation [47]. Conversely, the distant regulatory elements show differential GpC accessibility—especially for *VTRNA1-1*—suggesting that epigenetic regulation of these regions could contribute to cell type-specific vault RNA expression [47]. Since vault RNA levels can change profoundly in response to starvation or viral infections, it will be important to decipher the contribution of the various proximal and distal polymerase III promoter elements on the transcriptional regulation of vault RNAs in these contexts.

## 4. Beyond the primary transcript—from modifications to processing

In addition to transcriptional regulation, post-transcriptional modifications and processing events contribute to the modulation of vault RNA abundance and function. Initial analyses revealed that the rat vault RNA is an uncapped RNA with a 5′ triphosphate (pppG) and an intact 3′ poly-uridine track [2,17]. It has been proposed that DUSP11-mediated dephosphorylation of the 5′ pppG promotes processing and turnover of vault RNA transcripts [48,49]. Accordingly, vault RNA levels are increased in DUSP11 knockout (KO) cells [49]. Interestingly, an infection-dependent reduction in DUSP11 levels results in the accumulation of 5′ pppG vault RNAs that was proposed to trigger an innate immune response via RIG-I like receptors [50]. In addition, uridylation and subsequent binding of the exoribonuclease DIS3L2 to vault RNAs serves as a 3′ directed cytoplasmic quality control and degradation mechanism [51,52].

The most prominently studied modification of vault RNAs is the NSUN2-dependent deposition of 5-methylcytosine (m$^5$C; [53–55]. The m$^5$C modification at C69 of vtRNA1-1 has been demonstrated to regulate vtRNA1-1 processing into smaller regulatory fragments [53,55]. These so-called svRNAs (small-vault RNAs) derive from the primary vtRNA1-1 stem region in a Dicer-dependent way and regulate their target genes (e.g. CYP3A4 and CACNG7/8) in a miRNA-like fashion [53,56] (reviewed in [57]). The abundance of svRNAs 1, 2 and 3 was shown to increase upon NSUN2 depletion and in models of multi-drug resistance, while their relevance under physiological conditions remains unclear [42,56,58]. By contrast, the levels of svRNA4—whose 5′ end starts with the modified C69—decreased upon NSUN2 depletion associated with increased binding of SRSF2 to the primary vtRNA1-1 transcript [55]. The m$^5$C modification and resulting processing into svRNA4 were further proposed to be altered during cell differentiation [55].

Other vault RNA modifications include N$^6$-methyladenosine (m$^6$A [59]) and pseudouridylation (Ψ [60]). It is tempting to speculate that the high similarity between the internal promoter elements of vault and tRNAs genes could result in common RNA processing and modification events. However, a systematic approach to either is pending.

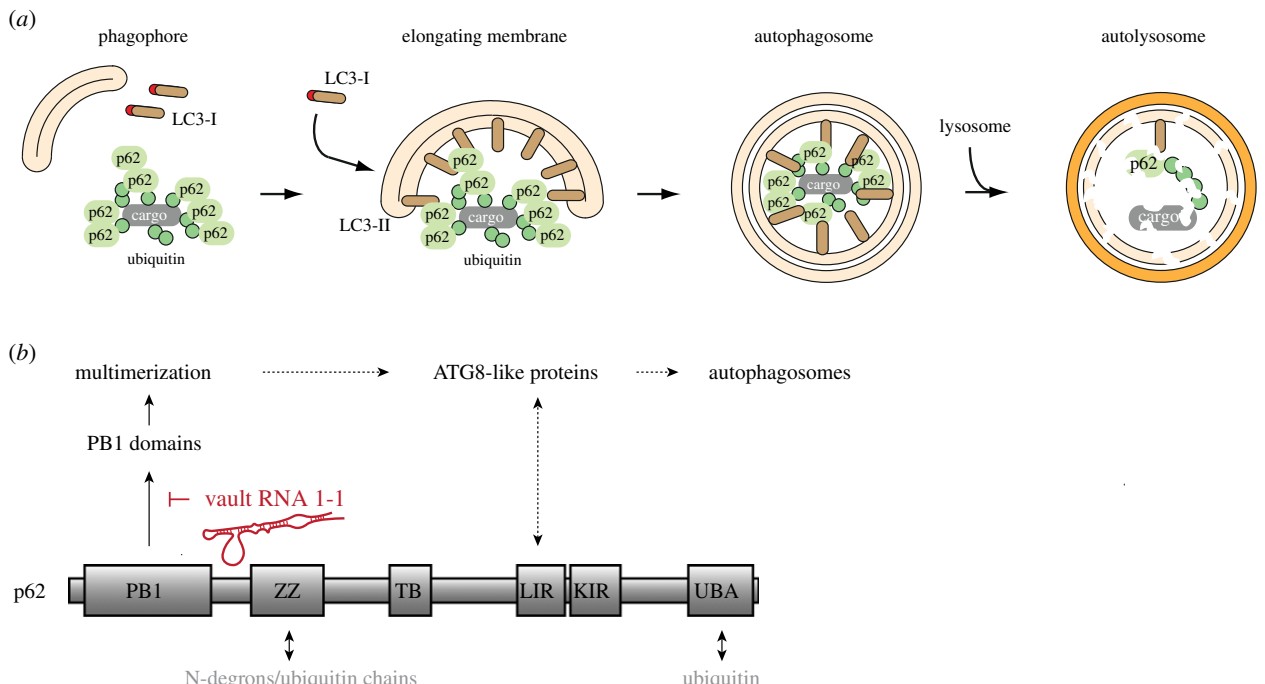

**Figure 2.** p62 and its function in selective autophagy and aggregate clearance. (*a*) Schematic overview of p62-dependent selective autophagy and aggregate clearance. P62 directs intracellular cargo to the phagophore. A double membrane vesicle is formed that fuses with the lysosome to degrade its content. (*b*) Schematic overview of the p62 domain organization. Cargo-binding sites are indicated below; vtRNA1-1 mediated regulation of p62 effector function is depicted above.

# 5. Vault RNA structural features

All vault RNAs identified to date are predicted to form distinct stem-loop structures in thermodynamic models [2,11]. RNase H structural probing of cell-derived human vtRNA1-1, vtRNA2-1 or mouse mVR1/VAULTRC5, however, suggests a far more open conformation of the central loop region (figure 1) [20,61]. In line with this, mutational analysis of the mouse vault RNA revealed increased affinity to Tep1 upon destabilization of existing complementarities within the central loop region (G70A, C73U) [61], indicating that a single-stranded nature of the central loop region could favour protein binding. Still, the tertiary structure of vault RNAs remains to be determined.

# 6. Unleashing non-coding power in autophagy—vault RNA 'riboregulates' p62

Since its discovery more than thirty years ago, the vault particle has been functionally implicated in drug resistance, apoptosis and nuclear transport (extensively reviewed in [62]). Only few studies focused on vault RNAs as an entity separate from the particle, although the large majority of these transcripts is not associated with it [17]. These studies suggested a role for vault RNAs in viral defence, apoptosis and multi-drug resistance [19,20,39,40,63–65]. Moreover, human vault RNAs can be constituents of microvesicles [66], and a recently identified vault RNA in *Trypanosoma brucei* has been implicated in mRNA trans-splicing [32]. However, a detailed understanding of their molecular functions and mechanism of action remained to be uncovered.

We recently discovered that vtRNA1-1 binds to and regulates p62-dependent autophagy and aggregate clearance,

thereby unravelling a first direct function of this conserved non-coding RNA [28]. Macroautophagy—further referred to as autophagy—is an essential cellular process that entails tethering, degradation and recycling of intracellular cargos including protein aggregates, excess or damaged organelles as well as pathogens. Thereby, autophagy fulfils a key function in cellular homeostasis and provides recycled material as a resource for the anabolic needs of the cell. During the process, cargo is selected and enclosed in double membrane vesicles called autophagosomes which eventually fuse with lysosomes leading to content degradation and release of recycled amino acids, lipids and nucleosides (reviewed in [67–69]).

# 7. Oligomerization is a key to p62 function in autophagy

The selectivity of autophagic processes is governed by autophagic receptors such as p62. These discriminate between different substrates by the usage of their cargo recognition domains and associate with autophagosomal membranes via a separate domain, thereby targeting intracellular cargo towards growing phagophores (figure 2) [70–74]. p62 plays a pivotal role in the autophagic clearance of intracellular cargos that fail to undergo degradation via the ubiquitin–proteasome system [75–79]. These include aggregation-prone cargos that are difficult to unfold and/or exceed the proteasomal capacity due to their mere size or quantity. As a result, p62 is of paramount importance in situations of acute proteotoxic stress and starvation [80]. Interestingly, p62-dependent assemblies have further been suggested to trigger autophagosome formation, implying a regulatory role in autophagy that was not previously anticipated [75,76,81].

p62 associates with ubiquitin via its C-terminal ubiquitin-associated domain (UBA) and binds to N-arginylated

royalsocietypublishing.org/journal/rsob   Open Biol. 10: 190307

**Figure 3.** Riboregulation of p62-dependent autophagy and aggregate clearance by vault RNA 1-1. We propose vtRNA1-1 as negative modulator of autophagy that directly binds to p62 and thereby inhibits p62 oligomerization, a prerequisite for the receptor's involvement in autophagy and aggregate clearance. The vtRNA1-1 to p62 ratio is thereby determining the extend of riboregulation. Conditions that are known to change RNA or protein levels are indicated in red and blue, respectively.

peptides as well as ubiquitin chains via the ZZ-type zinc (ZZ) finger domain (figure 2b) [75,82]. In turn, the LC3-interacting region directly binds to Atg8-like proteins such as LC3 and GABARAP, and thereby facilitates elongation of the autophagic membrane [73,83,84]. Since the intrinsic affinity of p62 for its cargos as well as Atg8-like proteins is rather weak, p62 multimerization is essential to achieving high avidity interactions while maintaining selectivity [84,85]. The interaction of p62 with itself and other autophagic receptors is mediated by the N-terminal Phox and Bem1 (PB1) domain (figure 2b) [71,86]. Strikingly, p62 PB1 domain mutants that are oligomerization-deficient fail to engage in autophagy [87], further emphasizing the importance of oligomerization for the 'effector' function of p62. The N-terminal linker region between the PB1 and ZZ domains (figure 2) was recently shown to contribute to p62 oligomerization and has been suggested to play an autoregulatory function [71,88]. The ZZ domain of p62 complexes two zinc atoms involving its conserved 4 Cys and 2 Cys-2 His motifs, and folds into a cross-brace zinc finger as also found for other ZZ-type domains [89]. It accommodates a negatively charged surface patch that serves as a binding site for N-arginylated peptides—so-called N-degrons—and synthetic autophagy inducing ligands (figure 2b) [75,89]. In addition, the ZZ domain has been reported to bind to K48- and K63-linked but not linear ubiquitin chains [82]. This intermolecular cross-linking of p62 via poly-ubiquitinated substrates [82], and conformational changes upon cargo binding to the ZZ domain have been reported to support p62 multimerization [75,76]. Yet, the regulation of p62 oligomerization—a key to its 'effector' function—is far from being fully understood.

We have recently shown that binding of vtRNA1-1 to p62 inhibits the receptor's oligomerization and engagement in autophagy. Mutagenesis of residues R139 and K141 to alanine in the ZZ domain of p62 reduced the interaction with vault RNA1-1. These residues are part of a positive surface patch in close proximity to the N-degron-binding site. Interestingly, we also observed decreased vtRNA1-1 binding to the p62 oligomerization mutant R21A/D68A/D73A (PB1 m) [28]. These data indicate that the both, the ZZ and the PB1 domain, are involved in mediating p62's RNA-binding activity. The involvement of other p62 domains in RNA binding has, however, not yet been excluded. Structural studies of p62 with and without RNA will help to decipher the complex interplay between the autophagy receptor and its 'riboregulator'. It further remains to be investigated whether vtRNA1-1 inhibits oligomerization of p62 independent of cargo binding, or whether the RNA prevents cargo association or influences cargo specificity, thereby inhibiting oligomerization.

## 8. 'Riboregulation' of autophagy—a response to different cues?

Vault RNA levels decrease during starvation [28] and increase profoundly upon different viral infections [19,20,39,40]. In detail, vtRNA1-1 levels diminish about two-fold in HuH-7 cells cultured in a minimal medium lacking amino acids and serum [28], whereas the levels of the other vault RNA paralogues remain more stable. The drop in vtRNA1-1 abundance was unaffected by p62 depletion and by treatment with bafilomycin A1, an inhibitor of the proton pump V-ATPase and hence of autophagosome-to-lysosome fusion. These data indicate that the decrease is not a result of RNA co-degradation with p62 during autophagy. Moreover, the starvation-induced decrease of vtRNA1-1 that was ectopically expressed from a heterologous promoter (H1), implies that the regulation of transcript levels is mediated by features present within the transcribed vtRNA1-1 gene body. Still, the detailed mechanism of this regulation remains to be determined.

The reduction of vtRNA1-1 expression during starvation is associated with a decrease in vtRNA1-1 binding to p62 [28]. This decreased RNA binding in turn promotes p62 oligomerization and autophagy, as discussed above. Treatment with bafilomycin A1 resulted in the accumulation of readily formed autophagosomes inside the cell, including enclosed cargo and autophagic receptors. Interestingly, while p62 accumulated upon bafilomycin A1 treatment, the amount of RNA-bound p62 remained constant. This observation further indicates that p62 which is actively involved in autophagy is free of vault RNA. Overall, the available data converge on a model where vtRNA1-1 levels are controlled by starvation and inhibit p62-dependent autophagy (figure 3).

Recently, ZZ domain-specific ligands were shown to potently induce p62-dependent autophagy [75]. VtRNA1-1 KO cells treated with the synthetic p62-ZZ domain ligand XIE62-1004-A activated autophagy significantly stronger than the respective CRISPR control cell lines [28]. This finding further affirmed the role of vtRNA1-1 as a negative 'riboregulator' of autophagy via p62 and the specificity of this modulation. Since p62-dependent autophagy can be initiated in multiple ways (reviewed in [68]), vtRNA1-1 mediated riboregulation could provide a mechanism to prevent overshooting autophagic activities.

By contrast to starvation, viral infections induce vault RNA expression. This response was observed in human cell culture models that were infected with members of the γ-herpesviridae family, including Epstein–Barr virus (EBV/HHV4) and Kaposi's sarcoma-associated herpesvirus (KSHV/HHV8), alpha-herpesvirus (Herpes simplex virus 1; HSV1) or paramyxovirus (Sendai virus (SeV) [19,20,39,40]. A similar increase was seen with human cell lines and mouse lung cells upon influenza A virus exposure [40]. Interestingly, most of these viruses are known to negatively modulate the autophagic flux of their host cells [90].

The transcriptional induction of vault RNAs upon infection has been associated with the expression of latent membrane protein 1 (LMP1) for EBV [39] and non-structural protein NS1 of influenza virus [40], respectively. In both cases high expression levels of vtRNA1-1 fostered an increase in viral load, while prior reduction of cellular vtRNA1-1 levels diminished viral replication *in vitro* and *in vivo*. Strikingly, Amort *et al*. [39] showed that the effect of vtRNA1-1 on virus replication is (i) concentration dependent, (ii) involving the vtRNA1-1 central domain and (iii) independent of MVP. These observations are well in line with the model of p62 riboregulation by vtRNA1-1.

Viruses are known to hijack key regulatory mechanisms of the cell to maximize viral replication while inhibiting cellular defence mechanisms. Upregulation of vtRNA1-1 levels could therefore serve to escape targeted viral degradation via autophagy and subsequent MHC class II antigen presentation [91]. In addition, it might force the cell to enter an anabolic, pro-proliferative state that can be exploited for rapid virus replication and to counteract cellular suicide programmes [92]. Therefore, overexpression of vtRNA1-1 during viral infection and the resulting deregulation of autophagy could serve in multiple ways to turn the host cell into a virus 'factory' while preventing an immune response (figure 3). Yet, the role of vtRNA1-1 mediated inhibition of autophagy in the context of viral infections has not been explored in much detail.

Furthermore, the molecular pathways that mediate apoptosis resistance in this setting remain to be elucidated [39]. Since p62 is involved in the crosstalk between autophagy and apoptosis (reviewed in [93,94]), increased levels of p62 upon autophagy inhibition via vault RNA 1-1 could contribute to modulate this crosstalk. In addition, the binding of vault RNAs and especially of vault RNA2-1 to protein kinase R (PKR) has been suggested to supress PKR activation upon influenza A infection and induce subsequent antiviral interferon response [40]. It will be interesting to uncover the function of the different vault RNA paralogues and their role in viral infections, as well as other physiological and pathological settings.

p62 levels decrease during autophagy due to phagolysosomal degradation [67] and increase upon proteasome inhibition [95–97]. Such changes in p62 levels could affect its relative ratio to vtRNA1-1 and influence the riboregulation of autophagy (figure 3).

The intracellular levels of p62 can be used as a marker to assess the autophagic state of the cell [67]. As mentioned above, p62 targeted to the autophagosome is ultimately degraded following autophagosome-to-lysosome fusion. In line with this model, we could show that autophagy-engaged p62 is not bound by vtRNA1-1 and consequently does not mediate autophagosomal degradation of the RNA [28]. Nevertheless, the levels of both p62 and vtRNA1-1 decrease

upon starvation, suggesting regulatory feedback that will be interesting to unravel.

In contrast with the starvation-induced decrease in p62 levels, proteasome inhibition has been reported to stimulate the Nrf1-dependent transcription of p62 [96]. This stimulation was shown to be essential for cellular survival during proteasome inhibition. Provided the increased levels of p62 shift the vtRNA1-1 to p62 ratio, proteasome inhibition could lead to inefficient vtRNA1-1-mediated riboregulation. This in turn would ensure a rapid sequestration of ubiquitinated proteins into 'sequestosomes' and allow proteasome-independent aggregate clearance via p62. Indeed, a significantly higher clearance of protein aggregates was observed in vault RNA 1-1 KO cells upon proteasome inhibition compared to the respective control cell lines [28]. This response was p62-dependent, since the expression of the p62 mutant S407A that is unresponsive to ULK1-dependent phosphorylation and the activation of the UBA domain in this context [98], abrogated the effect. Moreover, no difference in vault RNA levels was observed upon proteasome inhibition (R.H. 2017–2020, unpublished observations).

# 9. The new vaultAge—perspectives

The riboregulation of autophagy by a small non-coding RNA raises several additional perspectives, discussed below.

## 9.1. Identification of factors that control p62-vtRNA1-1 riboregulation

Identification of p62 and vtRNA1-1 as a riboregulatory effector pair calls for the identification of those cellular and/or viral factors that determine the abundance of both components as well as their interaction. It will be of great interest to assess transcriptional, post-transcriptional and—in the case of p62—translational events that influence intracellular p62 and vault RNA1-1 levels. In addition, post-translational and epi-transcriptomic RNA modifications, respectively, could represent a fast way to control the RNA–protein interaction. While modifications within the binding interface could directly affect ribonucleoprotein complex formation, others that influence conformation or localization could indirectly contribute to the modulation of this interaction.

Vault RNAs modifications have been recently described (see above), and m$^5$C has been functionally implicated in cell differentiation [53,55]. Yet, whether and to what extent these modifications can affect the riboregulation of autophagy remains to be investigated. Conversely, several post-translational modifications are known for p62 [99]. These include the LRRK2-dependent phosphorylation of T138 [100] within the ZZ domain [28]. Interestingly, LRRK2 is involved in the regulation of p62-depenent autophagy, and it has been discussed as a druggable target for Parkinson's disease [101]. The assessment of further modifications that might influence vtRNA binding to p62 will be of great interest, possibly also with regard to designing compounds that exploit the mechanism of 'riboregulation'.

## 9.2. Biological scope of p62 riboregulation by vtRNA1-1

It will be interesting to systematically determine the levels of p62 and vault RNA 1-1, and assess their interaction in cellular

royalsocietypublishing.org/journal/rsob    Open Biol. **10**: 190307

processes and stress conditions that are highly dependent on autophagy and its clearing function. These include differentiation and development [102], cell-death signalling [103] and cell-cycle regulation [104]. For example, the levels of non-coding RNAs including vault RNAs have been reported to be highly regulated during cellular differentiation [105]. Riboregulation could therefore represent a mechanism to modulate p62 function in this context. Thorough the data-mining of existing multi-omics, datasets might be an effective way for a first assessment of this possibility.

In addition, p62 plays a pivotal role as a downstream effector for various other cellular pathways (reviewed in [93,94]). While p62 has been described to influence mTORC1 activation in response to amino acids [106,107], we did not observe changes in the phosphorylation of the canonical mTORC1 targets ULK1 and 4E-BP1 upon vault RNA 1-1 depletion [28].

## 9.3. The role of p62 riboregulation in human disease

p62-dependent autophagy and aggregate clearance has been associated with neurodegenerative diseases, bone disorders and cancer [68,99,108]. Consequently, it will be of interest to explore whether vault RNA1-1-mediated riboregulation is altered under these pathological conditions and whether it could serve as a target for drug development.

Quality control and immunomodulatory functions of autophagy are considered to be tumour suppressive at early stages, while promotion of cellular growth and resistance to stress and drugs are believed to be tumour-promoting facets of autophagy in advanced tumours [93,95,108–110]. While the vault particle and vault RNAs have been previously associated with multi-drug resistance [7,56,63–65,111,112], the functional and molecular links between autophagy and multi-drug resistance are highly influenced by their biological contexts. Therefore, it remains an open question to what extent the riboregulation of p62 oligomerization by vtRNA1-1 and the subsequent modulation of p62-dependent autophagy and aggregate clearance are relevant in these pathological settings.

## 9.4. The role of the vault particle and MVP in p62-dependent aggregate clearance

Previous work gave no positive indication of the involvement of MVP in the riboregulation of autophagy [28] or in apoptosis resistance upon EBV infection [39], both of which are mediated by vault RNA. With the newly identified role of vtRNA1-1 in autophagy, it will be interesting to see whether the vault particle or MVP are directly or indirectly connected with this cellular process.

## 9.5. How widespread is riboregulation?

Work on the vtRNA1-1-p62 interaction uncovered an example of riboregulation, a process in which a protein's function is controlled post-translationally by the direct binding of a regulatory RNA [28]. Earlier examples of this emerging biological principle include 6S RNA regulation of RNA polymerase activity in bacteria [113], and the activation of immune receptors or the eIF2$\alpha$ kinase PKR by viral RNAs in mammalian cells [114,115] (reviewed in [116]). With hundreds of recently discovered RNA-binding proteins (RBPs) [117] involved in key cellular functions, we predict the widespread occurrence of riboregulation in the control of cellular processes. Characteristic features of riboregulation include (i) the direct interaction between the regulatory RNA and its target protein, (ii) the regulation of the levels and/or activity of one or both binding components by a biological cue and (iii) a change of the protein's function (in the widest sense) caused by RNA binding. Riboregulation of an RBP by an RNA is distinct from previously described examples of 'moonlighting', where several metabolic enzymes have been found to moonlight as RBPs and regulate the translation, stability or other aspects of the fate of RNAs [117,118].

To systematically survey for the biological scope of riboregulation, it will be informative to determine the RNA-binding proteomes under various physiological and pathological conditions [117,119–122]. Proteins displaying differential RNA binding can then be studied for their respective RNA-binding partners to uncover functional interactions [123,124]. Thus, the co-evolution of RNAs (including but not limited to non-coding RNAs) with proteins may have constituted a biological regulatory layer that was not previously anticipated.

Data accessibility. This article has no additional data.

Competing interests. We declare we have no competing interests.

Funding. We received no funding for this study.

Acknowledgements. We would like to recommend the websites devoted to the vault particle and its components that are maintained by the Rome Lab (https://vaults.arc.ucla.edu/pages/scientists) as well as the McManus Lab (https://mcmanuslab.ucsf.edu/node/256) and thank both labs for their maintenance. We acknowledge Ina Huppertz and Dmytro Dziuba for their helpful input and fruitful discussions.

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
