## [Reviewer comments · Open Biology]

Review History

RSOB-19-0307.R0 (Original submission)

Review form: Reviewer 1

Recommendation

Accept with minor revision (please list in comments)

Do you have any ethical concerns with this paper?

No

Comments to the Author

This is a very nicely drafted and well-structured review about the functional role(s) of the vault RNA (vtRNA) in various model systems and during different growth conditions. The vtRNA is a ~100 nt long ncRNA initially found to be associated to the vault complex which represents the largest known RNP in eukaryotic cells. Later it was uncovered that the majority of the vtRNA transcripts do actually not bind the vault complex and thus have a role beyond this gigantic RNP complex. This review by Büscher et al. focuses on these vault complex-unrelated functions of the vtRNAs. The main emphasis of the review is based on a recent publication by the authors group (Horos et al., 2019). In this publication they uncovered a regulatory role for one of the human vtRNA paralogs, called vtRNA1-1, in binding p62 and thereby negatively regulating autophagy in HuH-7 cells. The authors refer to this kind of physiological role as “riboregulation” which means that a ncRNA modulates the activity of an RNA-binding protein.

As mentioned above the submitted review manuscript is of high quality and cites most of the relevant literature. However, the manuscript can still be further improved by clarifying/addressing several points:

1) As a reader one gets the impression throughout the manuscript that riboregulation (as defined by the authors as ncRNA binding to a protein affecting the function of the polypeptide) is a novel and (almost) unprecedented finding. While Büscher et al. refer to 6S RNA-mediated regulation of RNA polymerase (Wassarman & Storz, 2000) and viral RNA regulating PKR (Meurs E, 1990), many more related "riboregulators" have been described in the past (see for example the review by Beckmann et al., 2016, DOI 10.1007/s00424-016-1819-4). In order to present a more balanced picture on the field's knowledge about "riboregulation" the authors might want to consider expanding on that in their submission.

2) page 4, line 78: a suitable reference for the work on pre-miR-886 should be given here as well (e.g. DOI: <https://doi.org/10.5808/GI.2015.13.2.26>)

3) for consistency reasons the vault RNA should be abbreviated as vtRNA (and not vRNA as on page 5, line 94)

4) page 6, lines 108-109: reference to (Li et al. 2015) is given twice here

5) page 9, last paragraph ("Unleashing..."): while I appreciate the authors previous work on the role of vtRNA1-1 on binding p62 and regulating autophagy in HuH-7 cells (Horos et al 2019), referring to other studies that have been published over the years on other physiological roles of vtRNA (or processing products thereof) should be given here in order to provide a more balanced overview. For example it has been shown recently that vtRNA can regulate trans-splicing (doi: 10.1074/jbc.RA119.008580), can directly bind chemotherapeutic drugs (doi:10.1093/nar/gki809), or serve as inter-cellular communication molecule via microvesicles (<http://dx.doi.org/10.4161/rna.25281>; www.pnas.org/cgi/doi/10.1073/pnas.1712108114).

6) please clarify: page 13, lines 301-305: the authors write that upon starvation, total levels of p62 increase while the amount of p62-bound vtRNA1-1 decreases. From Fig. 4B,C of the original publication (Horos et al., 2019) it does however appear that p62 levels remain unchanged under starvation conditions.

Decision letter (RSOB-19-0307.R0)

13-Jan-2020

Dear Dr Buescher,

We are pleased to inform you that your manuscript RSOB-19-0307 entitled "High VaultAge' - non-coding RNA control of autophagy" has been accepted by the Editor for publication in Open Biology. The reviewer has recommended publication, but also suggest some minor revisions to your manuscript. Therefore, we invite you to respond to the comments and revise your manuscript.

Please submit the revised version of your manuscript within 7 days. If you do not think you will be able to meet this date please let us know immediately and we can extend this deadline for you.

To revise your manuscript, log into <https://mc.manuscriptcentral.com/rsob> and enter your Author Centre, where you will find your manuscript title listed under "Manuscripts with

Decisions." Under "Actions," click on "Create a Revision." Your manuscript number has been appended to denote a revision.

- 1) A text file of the manuscript (doc, txt, rtf or tex), including the references, tables (including captions) and figure captions. Please remove any tracked changes from the text before submission. PDF files are not an accepted format for the "Main Document".
- 2) A separate electronic file of each figure (tiff, EPS or print-quality PDF preferred). The format should be produced directly from original creation package, or original software format. Please note that PowerPoint files are not accepted.
- 3) Electronic supplementary material: this should be contained in a separate file from the main text and meet our ESM criteria (see <https://royalsocietypublishing.org/rsob/for-authors>). All supplementary materials accompanying an accepted article will be treated as in their final form. They will be published alongside the paper on the journal website and posted on the online figshare repository. Files on figshare will be made available approximately one week before the accompanying article so that the supplementary material can be attributed a unique DOI.

Online supplementary material will also carry the title and description provided during submission, so please ensure these are accurate and informative. Note that the Royal Society will not edit or typeset supplementary material and it will be hosted as provided. Please ensure that the supplementary material includes the paper details (authors, title, journal name, article DOI). Your article DOI will be 10.1098/rsob.2016[last 4 digits of e.g. 10.1098/rsob.20160049].

- 4) A media summary: a short non-technical summary (up to 100 words) of the key findings/importance of your manuscript. Please try to write in simple English, avoid jargon, explain the importance of the topic, outline the main implications and describe why this topic is newsworthy.

Images

Data-Sharing

It is a condition of publication that data supporting your paper are made available. Data should be made available either in the electronic supplementary material or through an appropriate repository. Details of how to access data should be included in your paper. Please see <https://royalsocietypublishing.org/rsob/for-authors> for more details.

Data accessibility section

Sincerely,
The Open Biology Team
mailto:openbiology@royalsociety.org

Reviewer's Comments to Author:

Referee:

Comments to the Author(s)

This is a very nicely drafted and well-structured review about the functional role(s) of the vault RNA (vtRNA) in various model systems and during different growth conditions. The vtRNA is a ~100 nt long ncRNA initially found to be associated to the vault complex which represents the largest known RNP in eukaryotic cells. Later it was uncovered that the majority of the vtRNA transcripts do actually not bind the vault complex and thus have a role beyond this gigantic RNP complex. This review by Büscher et al. focuses on these vault complex-unrelated functions of the vtRNAs. The main emphasis of the review is based on a recent publication by the authors group (Horos et al., 2019). In this publication they uncovered a regulatory role for one of the human vtRNA paralogs, called vtRNA1-1, in binding p62 and thereby negatively regulating autophagy in HuH-7 cells. The authors refer to this kind of physiological role as “ribo-regulation” which means that a ncRNA modulates the activity of an RNA-binding protein.

As mentioned above the submitted review manuscript is of high quality and cites most of the relevant literature. However, the manuscript can still be further improved by clarifying/addressing several points:

1) As a reader one gets the impression throughout the manuscript that riboregulation (as defined by the authors as ncRNA binding to a protein affecting the function of the polypeptide) is a novel and (almost) unprecedented finding. While Büscher et al. refer to 6S RNA-mediated regulation of RNA polymerase (Wassarman & Storz, 2000) and viral RNA regulating PKR (Meurs E, 1990), many more related “ribo-regulators” have been described in the past (see for example the review by Beckmann et al., 2016, DOI 10.1007/s00424-016-1819-4). In order to present a more balanced picture on the field’s knowledge about “ribo-regulation” the authors might want to consider expanding on that in their submission.

2) page 4, line 78: a suitable reference for the work on pre-miR-886 should be given here as well (e.g. DOI: <https://doi.org/10.5808/GI.2015.13.2.26>)

3) for consistency reasons the vault RNA should be abbreviated as vtRNA (and not vRNA as on page 5, line 94)

4) page 6, lines 108-109: reference to (Li et al. 2015) is given twice here

5) page 9, last paragraph (“Unleashing...”): while I appreciate the authors previous work on the role of vtRNA1-1 on binding p62 and regulating autophagy in HuH-7 cells (Horos et al 2019), referring to other studies that have been published over the years on other physiological roles of vtRNA (or processing products thereof) should be given here in order to provide a more

balanced overview. For example it has been shown recently that vtRNA can regulate trans-splicing (doi: 10.1074/jbc.RA119.008580), can directly bind chemotherapeutic drugs (doi:10.1093/nar/gki809), or serve as inter-cellular communication molecule via microvesicles (<http://dx.doi.org/10.4161/rna.25281>; www.pnas.org/cgi/doi/10.1073/pnas.1712108114).

6) please clarify: page 13, lines 301-305: the authors write that upon starvation, total levels of p62 increase while the amount of p62-bound vtRNA1-1 decreases. From Fig. 4B,C of the original publication (Horos et al., 2019) it does however appear that p62 levels remain unchanged under starvation conditions.

Author's Response to Decision Letter for (RSOB-19-0307.R0)

See Appendix A.

Decision letter (RSOB-19-0307.R1)

23-Jan-2020

Dear Dr Buescher

We are pleased to inform you that your manuscript entitled "High VaultAge' - non-coding RNA control of autophagy" has been accepted by the Editor for publication in Open Biology.

Sincerely,
The Open Biology Team
<mailto:openbiology@royalsociety.org>

Appendix A

Response to referees:

We thank the reviewer and the editor for their constructive and helpful feedback on our manuscript RSOB-19-0307 entitled “‘High VaultAge’- non-coding RNA control of autophagy”. We have followed the reviewer’s suggestions and included additional references, as detailed below.

1) We describe the vtRNA1-1- p62 interaction as an example of ‘riboregulation’ – a process in which the direct interaction of an RNA with a protein changes the protein’s function. We differentiate ‘riboregulation’ from ‘moonlighting’ RNA-protein interactions that influence RNA fate (e.g. translation or stability of the bound RNAs). Examples of moonlighting include e.g. binding of the 3’UTR of IFN-gamma mRNA by GAPDH or the iron-regulated binding of IRE containing RNAs by IRP1/aconitase, which both influence the translation/stability of the bound RNAs. To our knowledge, only relatively few examples of riboregulation have been described in molecular detail thus far. These include 6S RNA regulation of RNA polymerase activity in bacteria (Wassarman and Storz, 2000), the activation of toll-like innate immune receptors (Kato et al. 2011) or PKR (Meurs et al. 1990). Following the reviewer’s suggestion, we have included the nice review by Beckmann et al., 2016 (see l. 455 ff), which further highlights RNA as a regulator of protein activity, including these examples. We have also better defined riboregulation versus moonlighting in paragraph v) of the ‘Perspective’ section.

2) We have included the recommended citation by Lee et al. 2015 in the paragraph ‘Expression of vault RNAs’ (l. 147).

3) We thank the reviewer for spotting the differences in abbreviations and suggesting consistency. The inconsistencies resulted from different historical uses, which we initially honoured, but we agree that consistency is important for clarity.

4) Again, we thank the reviewer for spotting this doubling. We have removed one of the citations.

5) We fully respect the reviewer’s wish to expand the paragraph “Unleashing non-coding power in autophagy – vault RNA ‘riboregulates’ p62” to include the suggested work. We have now re-cited the work by Gopinath et al. when mentioning relations to multi-drug resistance (l.211 ff) and elaborated on the more recent work by Shurtleff et al., 2017 and Kolev et al, 2019 in (l. 213 ff). The work on processing products of vault RNA including their physiological implications was already included in the originally submitted version in the paragraph “Beyond the primary transcript – from modifications to processing” (l.173 ff).

6) We thank the reviewer for this comment. The protein levels of p62 decrease in conditions of starvation due to an increase in autophagy (Horos et al., 2019 Fig. 4 B, comparing lanes 1, 3, 5, 7). However, when applying the small molecule inhibitor bafilomycin A1, p62 that is targeted for degradation is stabilised in readily formed autophagosomes (Horos et al., 2019 Fig. 4 B, comparing lanes 4, 6, 8), because bafilomycin A1 prevents the fusion between autophagosomes and lysosomes, thereby leading to the apparent ‘accumulation’ of p62 (comparing e.g. Horos et al., 2019 Fig. 4 B, lane 7 to lane 8) that we had described as an ‘increase’. We have attempted to clarify this.